# A Study of Greek Graviera Cheese by NMR-Based Metabolomics

**DOI:** 10.3390/molecules28145488

**Published:** 2023-07-18

**Authors:** Evangelia Ralli, Apostolos Spyros

**Affiliations:** NMR Laboratory, Department of Chemistry, University of Crete, Voutes Campus, 710 03 Heraklion, Crete, Greece; lilaralli@hotmail.com

**Keywords:** graviera cheese, NMR spectroscopy, metabolomics, fatty acids, metabolite profile, amino acids, multivariate analysis

## Abstract

Graviera is a very popular yellow hard cheese produced in mainland Greece and the Aegean islands, and in three PDO (protected denomination of origin) locations. Apart from geographic location, type of milk and production practices are also factors that affect cheese composition, and make this dairy product unique in taste and aroma. In this work, ^1^H nuclear magnetic resonance (NMR) spectroscopy in combination with chemometrics has been used to determine the metabolite profile (40 compounds) of graviera cheese produced in different geographic locations, with emphasis on cheeses produced on the island of Crete. Organic acids and amino acids were the main components quantified in the polar cheese fraction, while the fatty acid (FA) composition of the lipid fraction was also obtained. Analysis of variance (Anova) of the dataset showed that γ-aminobutyric acid (GABA), conjugated linoleic acids (CLA) and linoleic acid differentiate gravieras produced in different areas of Crete, and that the total amino acid content was higher in cheeses produced in eastern Crete. Targeted discriminant analysis models classified gravieras produced in mainland Greece, Cyclades and Crete based on differences in 1,2-diglycerides, sterols, GABA and FA composition. Targeted and untargeted orthogonal partial least squares discriminant analysis (OPLS-DA) models were capable of differentiating gravieras produced in the island of Crete and hold promise as the basis for the authentication of PDO graviera products.

## 1. Introduction

“Graviera” cheese is one of the most popular types of cheese consumed in Greece, along with “feta” cheese, and is produced by dairy farms located throughout the country. It is traditionally made from 80% ewe and 20% goat milk, However, graviera cheeses can be prepared also either solely from ewe, goat and cow milk or from a mixture of different milks. Although graviera is produced all over the country, only three regions are certified to produce graviera cheese with the protected designation of origin (PDO) label, as dictated by Greek Law and European Union legislation [1], namely, *Graviera Agraphon*, *Graviera Kritis* and *Graviera Naxou*. Graviera *Agraphon* and *Kritis* are produced using ewe or a mixture of ewe/goat milk, whereas Graviera *Naxou* is produced using cow or cow and ewe milk. According to PDO label rules, PDO Gravieras must contain only milk from animal farms within the geographical region indicated. 

During cheese production, three important biochemical processes take place: glucolysis, proteolysis and lipolysis, liberating in the cheese lactate, amino acids and glycerides/free fatty acids, respectively. These chemical compounds play a crucial role in determining the product’s quality since they contribute to and determine the flavour and aroma characteristics of cheese. Lactate can be metabolized by bacteria present in the cheese curd to products that may affect cheese texture by promoting eye formation (CO_2_) or by increasing pH and eventually softening the interior of camembert cheese (CO_2_, O_2_). Lactate metabolism products can also cause defects in cheese texture and flavour, such as “late gas blowing”, which refers to cracks formed in the cheese mass during ripening by butyrate and H_2_, a procedure accompanied by the development of off-flavours [2]. Proteolysis refers to cheese protein hydrolysis by proteinases, enzymes that can either originate from bacteria indigenous to milk that have survived pasteurization or were added during cheese-making, such as starter or non-starter lactic acid bacteria, rennet, etc. The process of proteolysis contributes to cheese flavour and texture by breaking down the protein network and releasing amino acids [3]. Lipolysis is the hydrolysis of triglycerides by hydrolases, producing free fatty acids and minor glycerides such as diglycerides or monoglycerides. Hydrolases are classified as lipases or esterases depending on the nature of the substrate, the length of FA chain and enzymatic kinetics. Free fatty acids, especially short- and intermediate-chain FA, are responsible for the characteristic flavour and aroma of cheese, depending on their perception threshold, concentration and pH [4].

There are a number of factors that can have a significant effect on cheese composition and quality. Herd diet, which is usually based on local flora, can affect milk composition and microbiological content; therefore, it is considered as one of the main factors that contribute to cheese amino acid and FA content. For example, Tzora et al. demonstrated that dietary treatment of sheep enriched with omega-3 fatty acids can affect the omega-3 FA content of cheese as well as the bacterial populations [5]. *Alpine Asiago* PDO cheeses produced with milk originating from pasture-fed cows were characterized by higher amounts of lysine, choline and 2,3-butanediol, indicating the effect of animal feed on cheese composition [6].

The development of cheese quality characteristics can also be influenced by the pedoclimatic conditions in the geographical region the cheese is produced. A number of studies have been devoted to the analysis of Greek graviera cheese obtained from different areas of production. Using ICP-MS, Danezis et al. determined the elemental profile of Greek graviera cheeses and classified them into nine geographical categories/regions [7]. Notably, Cretan gravieras were found to contain higher levels of praseodymium (Pr) and neodymium (Nd), a finding attributed to the soil and vegetation in the island of Crete being rich in these elements. Another study of the microbiological and physicochemical analysis of Greek graviera cheese showed that “*Graviera Kritis*” was characterized by lower pH values and amino acid content than “*Graviera Naxou*”, possibly due to different non-starter lactic acid bacteria (NSLAB) acidifying and proteolytic activity [8]. Analysis by means of SPME-GC-MS in the same study showed that although the two graviera cheeses had similar volatile organic compounds’ (VOCs) profiles, there were some compounds uniquely identified in each cheese label. Vatavali et al. analysed the physicochemical properties, mineral content, FA composition and VOC profile of graviera samples produced in six different regions of Greece [9] and then later expanded the work to include another five graviera-producing regions [10]. The statistical analysis of the combined analytical data set showed that gravieras from Naxos was the most clearly differentiated group of samples, presumably due to milk composition and geographical differences [10].

Furthermore, different types of cheese require different milk treatment, specific production processes and precise maturation conditions for the final product to occur. The lipid fraction of a large number of different Greek PDO cheeses were studied using GC-MS, and the collected data were used to study whether physicochemical properties and FA profiles could act as markers for PDO label, milk and cheese type discrimination [11]. In a study of the free fatty acid profile of traditional Greek cheese varieties, Georgala et al. reported that Cretan graviera cheese contained more propionic acid while *Kefalotyri* cheese had a higher acetic acid content [12].

^1^H NMR spectroscopy has proven to be an extremely useful tool in cheese analysis, contributing to the determination of the lipid fraction as well as the water-soluble metabolite content of different types of cheese [13,14,15]. The metabolite profile obtained by NMR spectroscopy was used successfully to study the ripening stages of the *Grana Padano* [16] and *Fiore Sardo* [17] cheeses from Italy. Samples of *Parmigiano Reggiano* cheese were successfully differentiated from other varieties of Grana cheese produced in Eastern Europe by means of NMR spectroscopy combined with multivariate analysis, despite the fact that the cheeses were at a different ripening stages [18]. Using data obtained from a variety of analytical techniques, including ^1^H NMR spectroscopy, Brescia et al. were able to discriminate PDO from PGI (protected geographical indication) samples of mozzarella cheese utilizing untargeted multivariate statistical analysis [19]. Likewise, the production site of *Asiago d’ Allevo* cheese samples was identified by analysing the ^1^H NMR spectra of their organic extracts using untargeted multivariate analysis [20]. It has also been demonstrated that European Emmental cheeses can be discriminated according to their geographical origin by high-resolution magic angle spinning (HR-MAS) NMR spectroscopy [21]. Samples of “*Mozzarella di bufala Campana*” produced in different sites, yet included in the PDO geographical region of the cheese, were discriminated by the same technique [22]. 

To our knowledge, the polar and lipid metabolite profile of Greek graviera cheese has not been studied using the analytical NMR spectroscopy methodological approaches described above for a variety of other cheeses. The aim of this work is thus firstly to characterize the full polar and apolar metabolite profile of Greek graviera cheese, produced in the mainland, Crete, and the Aegean islands, and to examine the ability of liquid phase ^1^H NMR spectroscopy in combination with statistical analysis to classify graviera cheese according to area of production, with emphasis on the authentication of Cretan graviera. 

## 2. Results and Discussion

### 2.1. Analysis of ^1^H NMR Spectra

#### 2.1.1. Polar Fraction

A representative ^1^H NMR spectrum of the aqueous extract of Greek graviera cheese is depicted in Figure 1. Thirty-two compounds were identified in the ^1^H NMR spectra of the aqueous extracts and their detailed peak assignments are listed in Table 1. Through the use of a suitable internal standard, these compounds were quantified in all the studied cheese samples by integrating their NMR spectra. Appendix A provides production details and meta data for all samples studied, while Appendix A summarizes the mean values of these polar metabolites as a function of graviera geographic origin. 

In general, the composition of Greek gravieras was found to be similar to those of other hard and semi-hard yellow cheeses [13,16,19,21,23]. The graviera polar metabolite profile contained the organic acids lactate (8–22 g/kg), citrate (0–1572 mg/kg), succinate (0–1340 mg/kg) and acetate (0–1238 mg/kg), in decreasing order of the average value. The acetate content of the gravieras determined in this study was in the range reported by Georgala et al. [12] for *Graviera Kritis* (727–1074 mg/kg). In a previous study, Dudley and Steele [24] reported that some nonstarter *Lactobacillus plantarum* strains are able to produce succinate by citrate catabolism in Cheddar cheese. Succinate was present in all but one graviera cheese product in this study, and at levels similar to those reported for Emmental cheese [25], while citrate was present in 43 out of 74 graviera cheese samples. The presence of citric acid in cheese is highly affected by the presence of certain lactic acid bacteria species, namely, *Streptococcus lactis* ssp. *diacetylactis* and *Leuconostoc* spp., that metabolize citrate to diacetyl [26]. In a recent study, Bozoudi et al. [8] identified leuconostocs as a significant NSLAB component of both *Graviera Naxou* and *Graviera Kritis* cheeses, indicating that differences in citrate levels measured in the present study could be partly attributed to varying counts of NSLAB leuconostocs in the graviera samples. Benzoic acid was the only aromatic acid identified in the graviera samples, ranging between 3 and 34 mg/kg of cheese. In a recent study, Yerlikaya et al. examined the formation of benzoic acid and its relationship to the microbial properties of traditional Turkish cheeses [27]. Benzoic acid levels in graviera measured in this study are comparable to those reported in ref. [27] for ripened *kashar* cheese, which is of a similar type (yellow, semi-hard cheese) to graviera. Although hard and ripened cheeses tend to have higher values of benzoic acid, which acts as a natural antimicrobial agent, the type and count of LAB is also very important in determining benzoic acid levels in cheese.

In the present study, 18 out of the 20 proteinogenic amino acids present in graviera were identified by NMR, lacking only cysteine (Cys) and histidine (His). Cys protons overlap heavily in the NMR spectrum of polar cheese extracts, and thus Cys has not been identified so far in any cheese using NMR [18,21]. Histidine has been identified by NMR in *Grana Padano* cheese [16] and mozzarella [19]; however, the assignments are not in complete agreement with values from the literature. Moatsou et al. have identified 19 of the 20 proteinogenic amino acids in fresh and aged Graviera Kritis, with proline (Pro) not being possible to measure due to limitations of the particular experimental methodology used (HPLC with derivatization) [28]. Combining these results with those of the present NMR study it can be concluded that all 20 proteinogenic free amino acids (FAAs) are present in graviera cheese. Since ion exchange chromatography, which is another analytical methodology used for FAA quantification, is not suitable for determining Trp and Cys in cheese (as per [29]), NMR can be very useful as a fast and efficient complementary methodology for FAA quantification in cheese. The amount of total free amino acids (TFAA) in the graviera samples varied significantly, ranging between 0.8 and 54 g/kg, with a mean value of 13 g/kg. Τhe variability in TFAA content may be attributed to the large variation of the graviera samples’ maturation stage (3–22 months, Appendix A), since aging correlates positively with amino acid production via protein hydrolysis [18,30], but also due to differences in NSLAB and LAB cultures used by different producers. Moatsou et al. showed that the addition of starters has a significant effect on the proteolysis of *Graviera Kritis* cheese in the early stages of maturation [28], while in a similar study, Michaelidou et al. showed that the addition of adjunct commercial cultures in the cheese curd could highly affect subsequent amino acid release during cheese proteolysis [31].

Some non-proteinogenic amino acids, namely ornithine and γ-aminobutyric acid (GABA), were also identified in the polar extracts of Greek gravieras. GABA is an L-glutamate metabolite that has been quantified at high concentrations (up to 7 g/kg) in Cheddar cheese [32], and at lower concentrations (<400 mg/kg) in 22 different Italian cheeses. In the present study, it was identified in 27 of 74 gravieras at levels up to 3000 mg/kg, indicating the presence of lactic acid bacteria demonstrating glutamate decarboxylase activity in these samples [32].

Ornithine, a product of arginase activity on arginine, was identified in 65 of 74 graviera samples, with the highest ornithine content determined in a Cretan graviera sample (2.2 g/kg). The lack of ornithine detection in four of the samples is probably due to it either being present at concentrations below the level of detection (LOD) or because it was converted to putrescine via ornithine decarboxylation by decarboxylase-positive bacteria, as putrescine was identified in five samples. However, it should be noted that ornithine decarboxylase-positive bacteria are not part of added starter cultures and thus their presence may be the result of contamination during cheesemaking, or storage [33]. These two amino acids, γ-aminobutyric acid and ornithine, have recently been the focus of studies regarding their bioactive functions and impact on health [34].

Tyramine, a decarboxylation product of tyrosine, was also identified in a large number of graviera samples, evidently due to the presence of decarboxylase-positive microflora in the cheese. It has been reported that the tyramine content of Dutch cheese was highly affected by storage time and storage temperature, so this biogenic amine may represent a helpful marker of the aging stage of cheeses. Cretan graviera was not found to contain any sugars, in agreement with data reported in the literature for similar types of hard yellow cheeses. 

#### 2.1.2. Lipid Fraction

The ^1^H NMR spectrum of the lipid extract of Greek graviera is depicted in Figure 2, with the spectral region between 5 and 6.5 ppm in expanded fashion, while the peak assignment of the apolar graviera components is listed in Table 2. The ^1^H NMR spectrum of graviera is very similar to those of other cheeses [14,15], and contains all the expected proton peaks of long- and medium-chain fatty acids, as well as peaks attributed to butyric acid, caproleic acid and the conjugated linoleic acid, and rumenic acid (CLA), which are FAs typically identified in animal-derived lipids. CLAs have been reported to be significantly increased in milk fat from cows grazing pasture compared to typical dairy diets; thus, CLAs in cheese could be related to animal feeding type [35]. Grazing has also been used as a tool for the increase in ω3 and polyunsaturated fatty acids (PUFA) in sheep, dairy buffalos, and dairy goats [36]. It is worth noting that although caproleic acid and CLAs have been identified by NMR in Greek bovine milk, only CLAs were reported in a recent GC-MS study of the FA profile of Greek protected designation of origin (PDO) cheeses, [11], while neither caproleic acid nor CLAs were reported in recent studies of the FA profile of Greek gravieras [10,37]. The CLA content of Greek cheeses and its evolution during aging has also been the focus of earlier analysis [38,39].

Finally, NMR spectroscopy also provides access to the glyceride profile of cheese, analytical data which are not available from GC-MS analyses of fatty acids. The characteristic peaks of 1,2-diglycerides (1,2-DG) were identified in the graviera cheese apolar extract spectra, indicating the activity of lipase enzymes that hydrolyse triglycerides, producing free fatty acids, diglycerides and monoglycerides (MG). Additionally, both 2-MG and 1-MG monoglyceride peaks were identified in several (18 of 74) cheese spectra, possibly as a result of factors irrelevant to cheese making, since no repetitive pattern concerning type of milk, area of production or maturation was observed.

The ^1^H NMR spectra of the lipid extracts were integrated and the data were used to calculate the fatty acid profile of the graviera samples as % moL of total FA according to a previously published methodology [40], with the relevant equations modified to include fatty acids specific to cheese lipids. Rumenic acid (CLA) and caproleic acid provide independent proton signals (G and H, respectively, letter codes in Table 2) that can be used to quantify them directly, while a combination of signals is needed for the quantification of the rest of the FA. For example, the percentage of ω-3 PUFA, mainly linolenic acid (LN), in the samples was calculated using the following equation:(1)$$\left[\mathrm{LN}\right]=\frac{E}{E+F+I}\times 100$$

Accordingly, butyric acid (BA), linoleic acid (LO), monounsaturated acids (MUFA) and saturated fatty acids (SFA) were calculated by the following adjusted equations:(2)$$\left[\mathrm{BA}\right]=\frac{I}{E+F+I}\times 100$$
(3)$$\left[\mathrm{LO}\right]=\frac{\left(A-\left(4\times \frac{E}{3}\right)\right)}{B}\times 100$$
(4)$$\left[\mathrm{MUFA}\right]=\frac{0.25\times \left(C-\left(4\times G\right)-\left(2\times H\right)-\left(4\times \frac{E}{3}\right)-\left(4\times \frac{A-\left(4\times \frac{E}{3}\right)}{2}\right)\right)}{\frac{B}{2}}\times 100$$
(5)$$\left[\mathrm{SFA}\right]=\left[\frac{F}{E+F+I}-\frac{\left[\mathrm{CLA}\right]}{100}-\frac{\left[\mathrm{LO}\right]}{100}-\frac{\left[\mathrm{MUFA}\right]}{100}\right]\times 100$$

The quantitative analytical data obtained for the lipid fraction of the graviera samples studied are also reported in Appendix A. 

### 2.2. Analysis of Variance of Graviera Composition

In order to examine compositional differences between gravieras produced in different geographical locations, the polar and apolar compositional data obtained by NMR spectroscopy were analysed by Anova. Focusing on gravieras produced in the island of Crete, Appendix A lists the relevant values of the factor F and the level of significance *p* for each metabolite. Several metabolites show statistically significant concentration differences at a *p* level <0.05, indicating a large variability in chemical composition is present already within the island of Crete, and some representative cases are depicted in Figure 3. The distribution of γ-aminobutyric acid according to geographic origin, for example, appears to be very interesting, as GABA is present in 11 of 19 gravieras from Rethymnon, while only 5/36 gravieras from the rest of the island were found to contain any. Recently, Tsafrakidou et al. [41] studied the phenotypic and genotypic variability of lactobacilli isolated from mature Graviera Kritis produced at two traditional dairies, concluding that lactobacilli are associated with their particular production dairy ecosystem. Since the production of increased GABA has been attributed to the presence of selected NSLAB [32], this difference in Rethymnon gravieras may be related to locally present LAB strains. 

It is worth noting that gravieras from eastern Crete (Heraklion, Lasithi) contain almost double the amount of FAA compared to western Crete, and relatively lower amounts of SFA. Focusing on fatty acids that have not been reported in the literature so far in detail, we observe that gravieras produced in Crete contain similar amounts of CLA regardless of origin within the island, while gravieras produced in Heraklion have the lowest amounts of caproleic acid and the highest amounts of linoleic acid.

Further analysis by Anova showed that metabolite profile differences were also present for samples produced in different Aegean islands (Naxos, Tinos, Lesvos, Ios) but the small number of samples (n = 2) available for each island only allows for qualitative analysis at present. Nevertheless, it should be pointed that the CLA content of gravieras produced in Naxos (0.51 ± 0.1%) and Tinos (0.54 ± 0.09%) from cow milk was found to be significantly lower than those produced from sheep/goat milk in Crete (0.92 ± 0.2%). The increased levels of CLA when sheep/goat milk is used for cheese-making are well documented in the literature and for a variety of cheeses [42]. To allow for a better comparison between wider geographical regions, gravieras produced in mainland Greece were grouped together, and compared with the Cretan and Aegean groups as a whole. The Anova analysis results, presented in Appendix A, demonstrate that statistically significant differences between mainland, Cretan and Aegean gravieras are present in the NMR metabolite profile, with the most important differences observed for CLA, GABA, 1,2-DG, sterols, and LN, as depicted in Figure 4. Cretan gravieras contain larger amounts of CLA and ω3-polyunsaturated fatty acids, but lower amounts of caproleic acid and 1,2-diglycerides, compared to Aegean and mainland gravieras. As with CLA, the lower ω3-PUFA content of the Aegean gravieras could be attributed to the use of cow milk for cheese preparation, which is naturally lower in ω3 than sheep milk [36]. 

### 2.3. Discriminant Analysis

As demonstrated by Anova in the previous section, there is a large within-region metabolite variability for gravieras produced within Crete and the Cyclades. Thus, in order to facilitate the multivariate analysis of the data set as a whole, samples were initially categorized in three broad geographical areas (Crete, Cyclades, mainland). First, a targeted approach was pursued, using OPLS-DA unsupervised multivariate analysis of the polar and apolar metabolite data of the graviera samples. The score plot of this OPLS-DA analysis is depicted in Figure 5, and it can be observed that the metabolite profiles of the gravieras are able to classify them according to broad geographical origin successfully. Gravieras from mainland Greece are more widely distributed than the two island origins, as expected due to the significantly larger geographical area covered under this group. Gravieras originating from Crete, which represent the larger sample group, show good clustering. The loadings plot of the OPLS-DA model in Figure 5, in which the variable importance parameter (VIP) is reflected in the size of the metabolite data points, shows that the most important metabolites for the observed classification were 1,2-DG and sterols for the Cyclades group, and GABA for the mainland group, while Cretan gravieras were differentiated mainly due their FA composition differences (CLAs and butyric acid). Τhe low predictive ability of this OPLS-DA model precludes its use for classifying unknown samples at the moment, but could be improved by increasing the presently low sampling size of gravieras from the mainland and Cyclades.

Since the majority of the graviera samples available in this study were produced in the island of Crete, it was interesting to explore whether an OPLS-DA model could be obtained from the NMR data that could classify the graviera samples based on a dual Cretan/non-Cretan origin. Such models would be very useful as a first step towards future efforts to construct chemical composition databases for the authentication of Cretan PDO gravieras. To pursue this goal, two different OPLS-DA models were constructed, one based on metabolite data (targeted), and one based on spectral bucketing of the whole NMR spectra (untargeted, polar plus apolar extract spectra), and Figure 6 depicts the respective score plots of these two exploratory OPLS-DA models. Both modelling approaches correctly classify the gravieras based on Cretan/non-Cretan origin and hold promise in establishing authentication models for Cretan graviera based on NMR metabolomics. The untargeted approach based on spectral bucketing has a higher discrimination ability, as reflected in the cross validation-ANOVA (CV-ANOVA) analysis of this model, depicted in Appendix A, with a CV-ANOVA *p* value of 2.9 × 10^−6^ and the respective permutation test depicted in Appendix A. It understood that non-Cretan gravieras sampling is under-represented in these models; thus, current work in our group is focusing on enlarging the mainland gravieras sampling space. The large variability in graviera chemical composition observed in this study emphasizes the importance of multiplatform approaches in geographic origin discrimination studies of foods in order to take advantage of not just the metabolome, but also the proteome and metalome of the product in study. 

In conclusion, this work represents the first attempt to map the polar and lipid metabolome of gravieras produced in Greece, with a focus on the island of Crete. The main FA profile of graviera was consistent with published results obtained by GC-MS analysis in the literature. Statistically significant differences in minor FA components (CLA, ω3-PUFA) and some polar cheese metabolites (γ-aminobutyric acid, lactate, total amino acids) were observed according to the production area, both within the island of Crete and among gravieras produced in the Aegean islands and the Greek mainland. The NMR metabolome variability of the gravieras was used to develop classification models for three main production origins (Crete, Aegean, mainland) could serve as the basis for the development of more advanced Cretan graviera authentication models based on extended omics datasets combining multiple analytical approaches.

## 3. Materials and Methods

### 3.1. Chemicals

The standard reagents deuterium oxide, D_2_O (99.9 atom % D, contains 0.05 wt % 3-trimethyl-silyl propionic-2.2.3.3-d_4_ acid sodium salt TMSP) and chloroform, CDCl_3_ (99.8% atom % D contains 0.03% (*v*/*v*) tetramethylsilane TMS) were obtained from Deutero GmbH, Kastellaun, Germany.

### 3.2. Samples

For this study, 74 hard cheese samples were used for NMR analysis, and were obtained either directly from local dairy farms or bought from local markets. A total of 53 cheese samples were produced in the island of Crete, while the remaining 21 were produced in northern Greece in the islands of Mitilini, Naxos, Tinos and Ios, as well as in Peloponnesus, Macedonia, Epirus, Thrace, Thessaly and Central Greece during the cheesemaking periods of 2012–2014 and 2017–2019. The maturation time of the cheese samples varied from 3 months (minimum period of maturation) up to 22 months. 

### 3.3. Sample Preparation: NMR Analysis

The graviera samples were kept frozen at −18 °C until analysis, and were prepared according to a published solid food NMR analysis protocol [43], with small alterations. In brief, for the extraction of polar graviera components 8–10 g of cheese were freeze dried at −50 °C overnight. After freeze drying, solids were weighed and grounded with a pestle and mortar under liquid nitrogen. An amount of 0.30 g of grounded sample was added in 3 Eppendorfs of 1.5 mL, total 0.90 gr for each sample, and mixed with 1 mL ultrapure water for the extraction of polar compounds. Eppendorfs were sealed with laboratory film, placed into ultrasound bath for 30 min and centrifuged at 13,148× *g* for 10 min. Of the 3 formed phases, the middle polar phase was removed and transferred to a glass flask. The extraction was repeated twice, three times in total. The supernatants were collected in the initial glass flask (one for each sample) and freeze dried overnight. When dried, the precipitate was weighed to determine the total extracted solids and the weighed amount of the solid was dissolved in 700 μL D_2_O-containing 0.05% TMSP in a glass vial. Vials were placed into an ultrasound bath for 30 min and, then, the extracts were filtered through glass wool directly into a 5mm NMR tube for spectra acquisition.

To extract non-polar components, 1 mL chloroform was added in each Eppendorf mentioned above. Eppendorfs were sealed with laboratory film, placed into an ultrasound bath for 30 min and centrifuged at 6708× *g* for 10 min. Two phases were formed, from which the non-polar liquid phase was carefully removed and transferred into a glass flask. The extraction was performed 3 times in total, the supernatants were collected in the initial glass flask (one for each sample) and placed under vacuum overnight. The remaining lipid fraction was dissolved in 700 μL CDCl_3_-containing 0.03% TMS and flasks were placed in an ultrasound bath for 30 min. After sonication, the extracts were filtered through glass wool directly into a 5mm NMR tube for spectrum acquisition.

1-D and 2-D ^1^H and ^13^C NMR spectra were obtained on a Bruker Avance-III-500 spectrometer operating at 500.137 MHz for proton and 125.75 MHz for carbon at 298K. The water signal was suppressed by pre-saturation, when necessary. The following conditions were used for the acquisition of 1-D ^1^H spectra: 64 k data points, 256 scans, 8 dummy scans, 6.45 s repetition time, spectral width of 12.016 ppm, line broadening of 0.3 Hz. For ^13^C dept135 spectra, the conditions were: 65,000 data points, 7168 scans, 4 dummy scans, repetition time 2.62 s, spectral width of 160 ppm, line broadening of 1.0 Hz. Phase and baseline correction were applied to all 1-D spectra and 2-D spectra where necessary. All spectra were processed by standard TopSpin Software (Bruker Corporation, Billerica, MA, USA, v3.1). Signal assignment was achieved by standard 2D NMR gradient spectroscopy (gCOSY, gHSQC, gHMBC, gTOCSY, gHSQC-TOCSY) experiments, the Chenomx NMR Suite software (8.02) and comparison with online NMR databases such as The Human Metabolome Database “https://hmdb.ca (accessed on 13 September 2022)”, and the Spectral Database for Organic Compounds, SDBS “https://sdbs.db.aist.go.jp (accessed on 15 September 2022)”.

### 3.4. Spectra Integration and Multivariate Statistical Analysis

For targeted analysis, the ^1^H NMR spectra of aqueous and chloroform extracts were integrated manually using TopSpin software and the option for automated baseline correction of the integrals was selected for better accuracy. 3-(Trimethylsilylpropio nic-2,2,3,3-d_4_ acid sodium salt was used as internal standard for the quantification of the polar graviera extracts. Representative LOQ for signals with S/N = 10 had 0.05 μmoL/g of cheese, and the relative standard deviation was calculated via replicant analysis to be <15% for the metabolites quantified. 

For untargeted analysis, the ^1^H NMR spectra of aqueous and chloroform extracts were bucketed using Amix Software (Bruker Corporation, Billerica, MA, USA, v3.9.14) and the data were imported to Simca Software (Sartorius AG, Göttingen, Germany, v13.02) to develop multivariate statistical analysis models (PCA, OPLS-DA). Internal cross validation of the OPLS-DA models was performed automatically by the software by repeatedly dividing the samples in seven random groups, developing a DA model using the six groups (training set) and confirming its validity with the last group (test set) until a stable model was produced. 

## Figures and Tables

**Figure 1 molecules-28-05488-f001:**
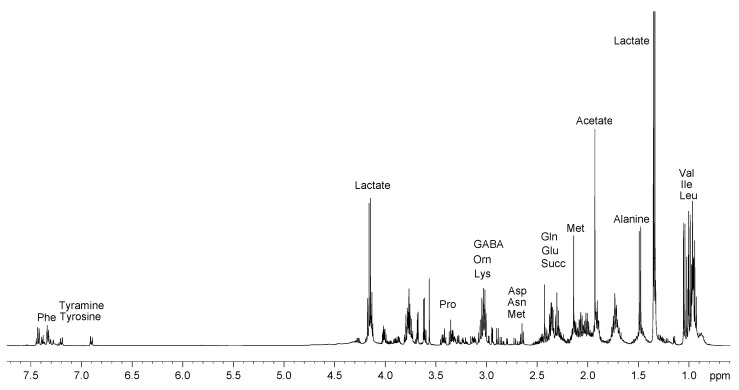
Typical ^1^H NMR spectrum of the water extract of graviera cheese sample in D_2_O-TSP, 500 MHz. Some metabolites are highlighted.

**Figure 2 molecules-28-05488-f002:**
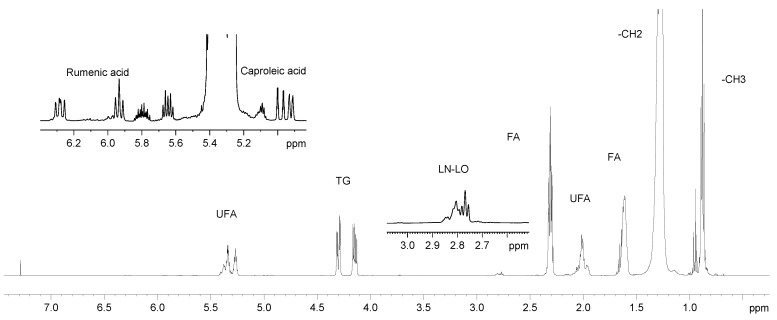
Typical ^1^H NMR spectrum of chloroform extract of graviera cheese sample in CDCl_3_ at 500 MHz. UFA: unsaturated fatty acids, TG: triglycerides, LN: linolenic acid, LO: linoleic acid, FA: fatty acids.

**Figure 3 molecules-28-05488-f003:**
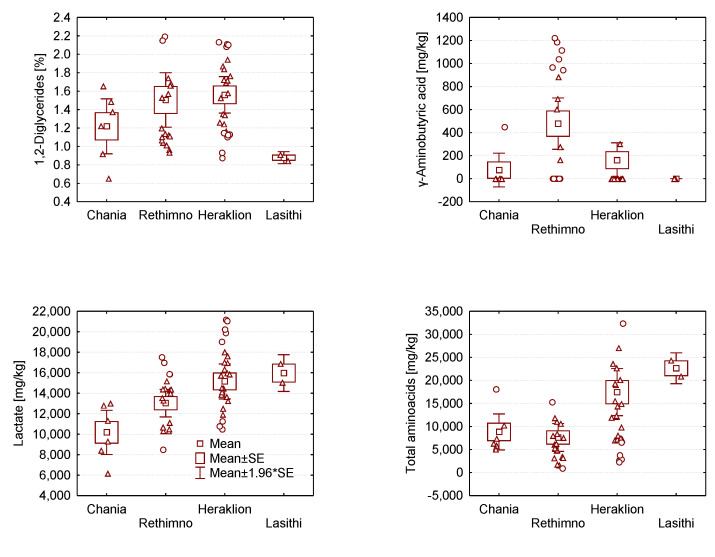
Box plots of 1,2-diglycerides, γ-aminobutyric acid, lactate and total amino acids (TFAA) in gravieras produced in Crete as a function of geographic area from west (Chania) to east (Lasithi).

**Figure 4 molecules-28-05488-f004:**
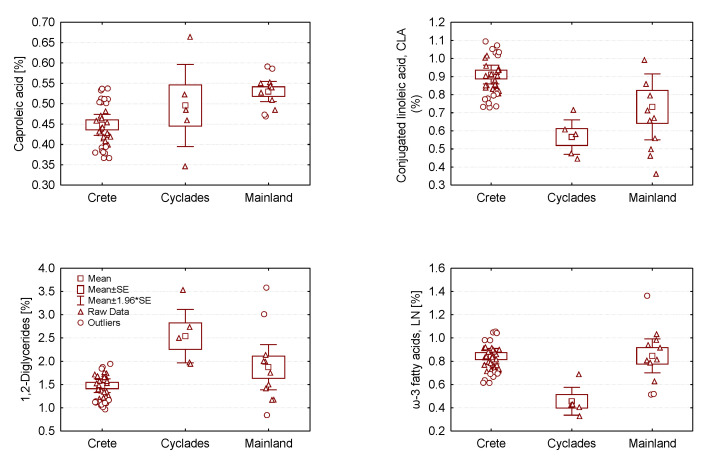
Box plots of caproleic acid, conjugated linoleic acid (CLA), 1,2-diglycerides and ω-3 polyunsaturated fatty acids (LN) in gravieras produced in Greece as a function of geographic area.

**Figure 5 molecules-28-05488-f005:**
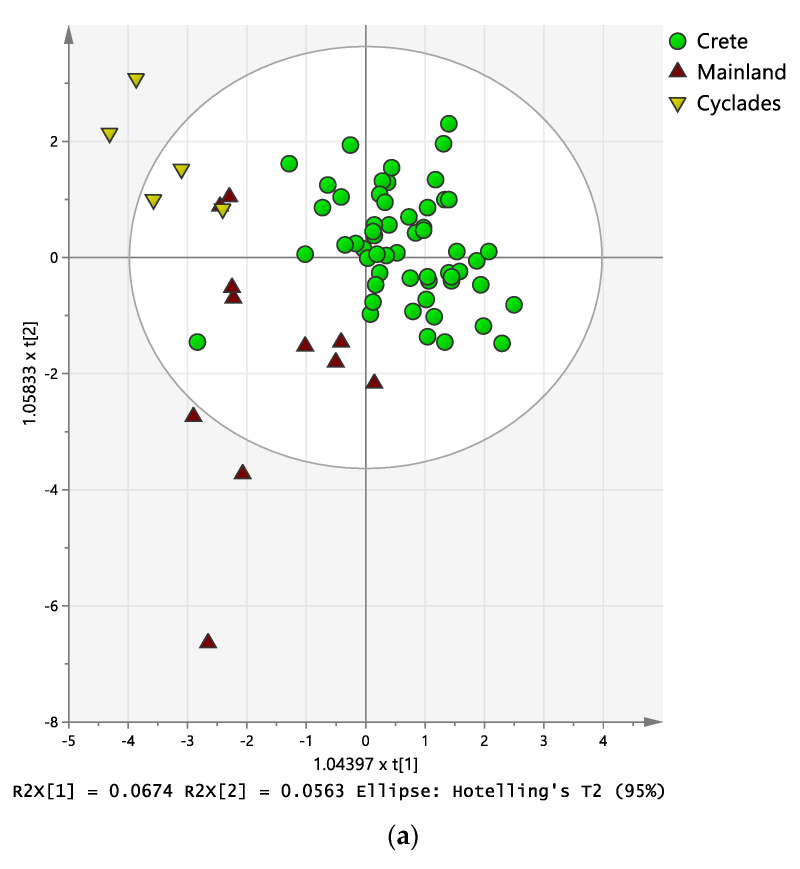
OPLS-DA model score plot of the first two predictive components, t1 and t2, (**a**) and scatter plot of loadings *p* and *q* (**b**) for the geographical origin of graviera samples based on their metabolite profile. R^2^X = 0.636, R^2^Y = 0.532, Q^2^ = 0.261. The size of metabolite symbols in the loading plot reflects their Variable importance Parameter (VIP) value in the model.

**Figure 6 molecules-28-05488-f006:**
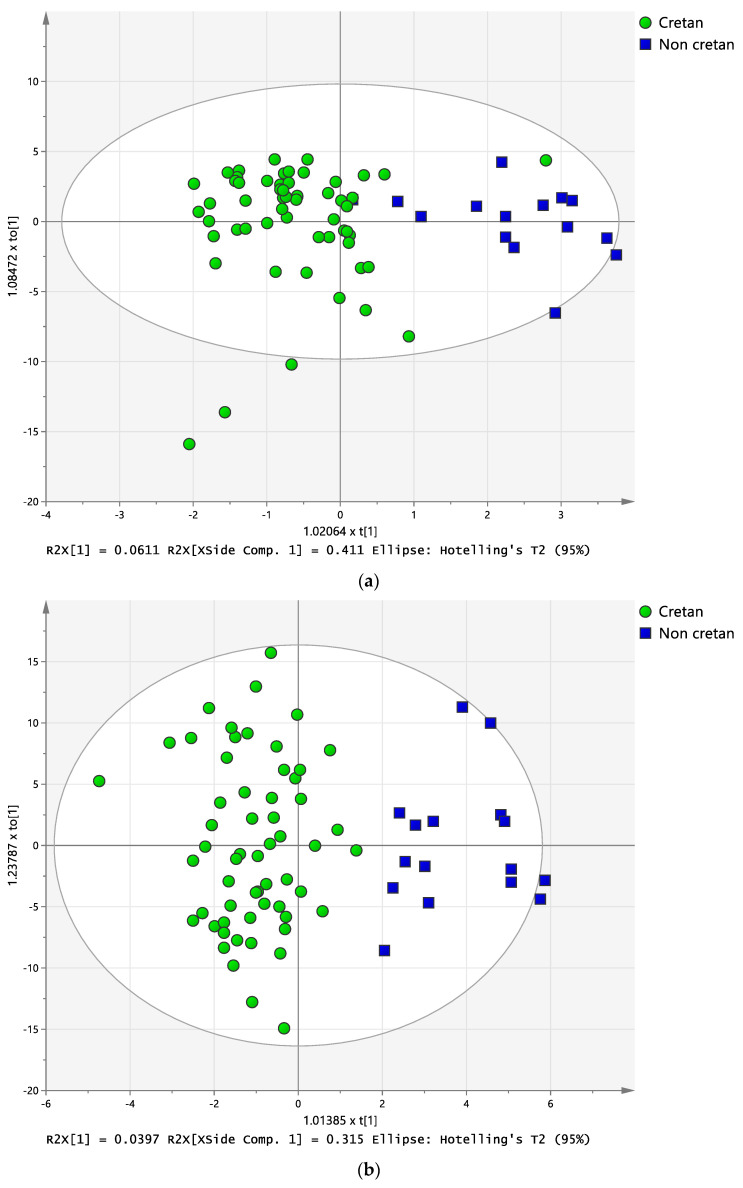
OPLS-DA model score plots of the predictive, t1, and the first orthogonal component, to1, from metabolite data (**a**) and NMR spectra buckets (**b**) for the differentiation of the origin of Greek gravieras (Cretan/non-Cretan).

**Table 1 molecules-28-05488-t001:** ^1^H and ^13^C NMR chemical shifts of the organic compounds identified in the polar extract of graviera cheese samples.

	Compound	Assignment	^1^H ppm	^13^C ppm
1	Acetate	α-CH_3_	1.92	26.1
2	Alanine	β-CH_3_	1.48	18.9
		α-CH	3.78	53.7
3	Arginine	γ-CH_2_	1.68	26.8
		β,β′-CH_2_	1.90	30.1
		δ,δ′-CH_2_	3.23	43.1
		α-CH	3.74	57.1
4	Asparagine	β-CH_2_	2.89	37.4
		α-CH	4.00	51.5
5	Aspartate	β′-CH_2_	2.80	39.0
		α-CH	3.90	53.6
6	Benzoate	2,6-CH	7.87	131.8
7	Choline	α-CH	3.21	56.7
8	Citrate	2,4-CH_2_	2.72/2.54	47.6
9	Cytosine	2-CH_2_	6.00	-
10	Formate	HCOO	8.50	173.5
11	GABA	β-CH_2_	1.91	26.4
		α-CH_2_	2.30	36.4
		γ-CH_2_	3.02	42.0
12	Glutamate	β,β′-CH_2_	2.09	30.1
		γ,γ′-CH_2_	2.36	36.3
		α-CH	3.76	57.2
13	Glutamine	β-CH_2_	2.14	28.9
		γ-CH_2_	2.46	33.6
14	Glycerol	2-CH	3.79	74.9
		1,3-CH_2_	3.65/3.56	65.4
15	Glycine	α-CH_2_	3.56	44.2
16	Isoleucine	δ-CH_3_	0.94	13.8
		β′-CH_3_	1.02	17.6
		γ′-CH	1.27	27.1
		γ-CH	1.46	27.1
		β-CH	1.99	38.5
		α-CH	3.68	62.4
17	Lactate	β-CH_3_	1.34	23.0
		α-CH	4.16	71.5
18	Leucine	δ,δ′-CH_3_	0.96	20.9/24.6
		β-CH2	1.72	42.5
		γ-CH	1.72	27.1
		α-CH	3.74	56.2
20	Lysine	γ-CH_2_	1.49	24.4
		β-CH_2_	1.72	29.2
		δ-CH_2_	1.90	33.1
		ε-CH_2_	3.03	41.9
		α-CH	3.75	57.1
21	Methionine	δ-CH_3_	2.14	16.7
		γ-CH_2_	2.65	31.5
		β-CH_2_	2.14	32.4
		α-CH	3.85	56.7
22	Ornithine	δ-CH_2_	3.06	39.3
		γ-CH_2_	1.77	25.5
		β-CH_2_	1.95	30.2
		α-CH	3.79	57.3
23	Phenylalanine	β-CH_2_	3.14/3.29	39.2
		α-CH_2_	4.00	58.9
		2,6-CH	7.33	131.8
		4-CH	7.37	131.9
		3,5-CH	7.43	132.1
24	Proline	3-CH_2_	2.01	27.3
		2-CH_2_	2.35	32.0
		4-CH	3.35	48.9
		4′-CH	3.41	48.9
		1-CH	4.14	64.1
25	Serine	α-CH	3.86	56.0
		β,β′-CH_2_	3.97	63.1
26	Succinate	2,3 CH_2_	2.41	36.2
27	Threonine	γ-CH_3_	1.33	19.5
		α-CH	3.62	62.2
		β-CH	4.27	66.0
28	Tryptophan	4-CH	7.73	118.0
29	Tyramine	α,β-CH_2_	2.93/3.27	34.8
		3,5-H	6.90	118.8
		2,6-H	7.22	133.7
30	Tyrosine	3,5-H	6.90	118.8
		2,6-H	7.19	133.7
31	Uracil	2-CH	5.80	103.8
		1-CH	7.54	146.3
32	Valine	γ-CH_3_	1.00	19.5
		γ′-CH_3_	1.05	20.7
		β-CH	2.28	31.9
		α-CH	3.63	63.2

**Table 2 molecules-28-05488-t002:** ^1^H NMR chemical shifts of the organic compounds identified in the non-polar extract of graviera cheese samples (lipids).

	Compound		Assignment	^1^H ppm	^13^C ppm	Letter Code
1	Sterols		–CH_3_	0.68	11.7	
2	FA except n-3/Butyric	ω1	–CH_3_	0.88	14.1	F
3	Butyric acid	H4	–CH_3_	0.94	13.8	I
4	ω-3 FA	ω1	–CH_3_	0.97	14.1	Ε
5	All FA		–(CH_2_)n-	1.26	29.4	
6	All FA	H3	–O–CO–CH_2_–CH_2_^−^	1.61	24.9	D
7	UFA-cis		–CH_2_–CH=CH–	1.97	32.4	C
8	UFA-trans		–CH_2_–CH=CH–	2.02	27.1	C
9	All FA	H2	–O–CO–CH_2_–CH_2_	2.30	34.0	B
10	PUFA (Linoleic)	H11	=CH–CH_2_–CH=	2.77	25.6	A
11	PUFA (Linolenic)	H11 H14	=CH–CH_2_–CH=	2.80	25.6	A
12	1,2-Diglycerides		HO–CH_2_–CH–	3.72	61.6	
13	1,3-Diglycerides		–CH_2_–O–CO–	3.98	64.3	
14	Triglycerides		–CH_2_–O–CO–	4.14	62.1	
15	Triglycerides		–CH_2_–O–OC–	4.30	62.1	
16	Caproleic acid	H10a	=CH	4.91	114.3	H
17	Caproleic acid	H10b	=CH	4.97	114.3	H
18	1,2-Diglycerides		–CH–O–CO–	5.09	72.0	
19	Triglycerides		–CH–O–CO	5.26	68.7	
20	UFA-cis		–CH=CH–	5.33	129.8	
21	UFA-trans		–CH=CH–	5.37	130.3	
22	CLA	H12	–CH=	5.63	134.6	
23	Caproleic acid	H9	–CH=CH_2_	5.78	139.0	
24	CLA	H10	–CH=	5.92	128.8	
25	CLA	H11	–CH=	6.27	125.6	G

## Data Availability

Data are contained within the article and the Appendix A.

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
