# Peer review of "A Study of Greek Graviera Cheese by NMR-Based Metabolomics"

_molecules, 2023, doi:10.3390/molecules28145488_

Round 1
Reviewer 1 Report
The research is well conducted, the experimental design has some flaws (e.g.: unbalanced samples populations) which have been anyway highlighted by the authors, data are well explained and the findings are of interest.
issues:
Some acronyms like NSLAB, VOC should be reported extensively at their first occurrence;
The authors redirect the reader to reff. 40 and 43 for sample extraction and preparation procedures: since those articles are not freely available, the authors should add a brief description of those procedures;
The authors should add the repetition time to the NMR parameters to ensure that the full relaxation condition has been achieved and the quantification from spectra integration is correct;
The regression parameters of the PLS model of maturation time (fig. 5) should be reported;
The OPLS-DA model is weak and has a very low predictive strength (R2Y=0.532, Q2=0.261). The authors should state this fact;
Concening Tyramine (lines 164-169), the authors suggest that “this biogenic amine may represent a helpful marker of the aging stage of cheeses.”. Since the authors obtained these data in the present research, they might perform the statistical analysis of the tyramine content as a function of the seasoning and add these results.
Generally good and understandable. Remove the repetition "detailed ... details" in lines 124-125
Reviewer 2 Report
Report on the manuscript molecules-2446508 entitled: A study of Greek graviera cheese by NMR-based metabolomics.
The authors have carried out good research, but I have some concerns regarding the structure and content of the manuscript. Specifically, the way that the results are presented and discussed must be improved.
There are already many studies on cheese metabolomics using NMR in the literature.
It is true whatever the authors say that the work offers complementary information on a specific variety of cheeses to create a database. Nevertheless, neither the year nor the maturation time that affects the concentrations of the metabolites considered have been considered, so the information provided is biased.
The discussion must be improved. With so much related literature, it is not acceptable only to mention whether the results agree or not with those already published. Why does or does not this happen? Discussion must be critical, not just descriptive and comparative.
The literature is full of tables just like Tables 1 and 2 of the manuscript. The information in Tables S3 and S4 is unique and specific to this study. Therefore, the authors should consider a combination of Tables 1 and 2 and S1 and S2 to be shown in the manuscript together with a proper critical discussion. That will improve the manuscript considerably. For example:
- L. 139. Why did this happen?
- L. 145-154. A very interesting fact but… why did this happen?
- L. 159-162. Same as above.
- …
- Please, improve the whole Results and Discussion section…
Other comments:
- Please, review the Suppl material because there are two “Table S1”. The last one must be “Table S5”.
- L. 54. “FA” abbreviation has not been previously defined.
- L. 133. “in decreasing order of the average value”.
--
Reviewer 3 Report
This manuscript deals with a study of a large panel of Greek graviera cheese, based on liquid 1H NMR extracts in deuterated water and chloroform.
After a clear and very instructive introduction, the authors present results and discussion. I found Figure 5 and the former paragraph, lines 266-279, as not enough conclusive, and could be removed. I do not agree with the conclusion that "the PLS model appears to cluster samples regardless of geographical origin": The groups are mixed.
There is also a lack of a general conclusion for the manuscript.
The main criticism I may formulate concern the Materials and methods part. There is no details on the preparation of samples. The authors should describe it, at least briefly, even if it is reported in Ref 43, a chapter book to which I do not have access. In addition, to is stated that "small alterations" occurred (line 345) which are not related. It would be useful to have experimental details about the state of the cheese when it undergoes extraction (ground?), which proportion of solvent vs cheese is used, how long was the extraction, was a thermodynamic equilibrium reached, at which temperature, what was the final pH of the water extract, and so on …
For the NMR analysis, neither the temperature nor the repetition time is reported. Repetition time is an important parameter for 1H 1D analysis, as quantitative data are obtained only if the repetition time is at least 5 times the longest T1 of signals. Which internal standard was used (line 123)? For quantification, extraction should be complete or at least reproducible. This should be specified.
I would like also more details about the assignment. Although the list of 2D NMR recorded spectra is given, the supporting Information may contain the observed correlations, and also the report of some 13C chemical shifts not reported in Table 1, but that could be obtained from gHMBC spectra. The detailed list of online databases to which experimental results were compared should also be reported.
Here are other minor points to be taken into account to help at reaching a high quality paper:
- Please explain each abbreviation at their first occurrence.
- Lines 143-144, which conclusion can be drawn from ref 28 in relation with the present study?
- Line 147, it would be more helpful to the reader to cite the two proteinogenic amino acids not found, than the eighteen that are found.
- Equations 1-4: brackets are more generally used for concentrations.
- Figure 3: Nicely the authors ordered areas from West to East, which could be mention.
- Figures 5 and 6: please provide more readable characters, and more understable captions of axes.
- Line 363: does "replicant analysis" means that sample preparation and NMR analysis were performed several time for each sample?
Overall, this work reports a valuable work that should be published, provided the authors reply to the raised points.
Round 2
Reviewer 2 Report
Report on the manuscript (R1) molecules-2446508 entitled: A study of Greek graviera cheese by NMR-based metabolomics.
The manuscript has improved considerably.
- Personally, I am not a fan of the term “type”. Phrases such as “milk type”, “cheese(s) type(s)” are not correct. (“type of milk/cheese… could be accepted).
Phrases such: “type(s) of cheese”, etc. I prefer “cheese variety” /variety of cheeses.
In addition, some specific examples:
L. 43. “camembert-type cheese” = Camembert cheese.
L. 99. “grana type” cheeses = from other varieties of Grana cheese….
- Some comments related to text structure and writing:
- Sometimes, the authors use “K” for “Kg” and others “k” (kg). Please, review.
- Please, use “.” Instead of “,” (comma) in the supplementary material Tables and Figures of the manuscript!
- Please, italicize the names of cheese varieties that are in Italian (or in a language other than English).
- Please, the description and use of abbreviations must be consistent. For example:
o L. 9. NMR not previously described.
o L. 54. The abbreviation “FA” is described as “free fatty acids”. Am I right? Then, why was it not used in L. 91 and 252. In fact, such abbreviation could be described in L. 31 and be used far more times…
o TFAA or FAA? L. 190, etc…
o TG, DG, UFA, FA (Figure 2)?
o Etc…
- Use the same number of decimals or significant figures in Table 1. Why do 13C (ppm) values have sometimes 1 and others 2 decimal digits?
- Table 1: some numbers must be subscripted.
Please, review the use of "type".
